# Current State-of-the-Art in Membrane Formation from Ultra-High Molecular Weight Polyethylene

**DOI:** 10.3390/membranes12111137

**Published:** 2022-11-12

**Authors:** Andrey Basko, Konstantin Pochivalov

**Affiliations:** G.A. Krestov Institute of Solution Chemistry, Russian Academy of Sciences, 153045 Ivanovo, Russia

**Keywords:** ultrahigh molecular weight polyethylene, membrane formation, powder sintering, thermally induced phase separation, phase diagram, structure formation, shrinkage of membranes

## Abstract

One of the materials that attracts attention as a potential material for membrane formation is ultrahigh molecular weight polyethylene (UHMWPE). One potential material for membrane formation is ultrahigh molecular weight polyethylene (UHMWPE). The present review summarizes the results of studies carried out over the last 30 years in the field of preparation, modification and structure and property control of membranes made from ultrahigh molecular weight polyethylene. The review also presents a classification of the methods of membrane formation from this polymer and analyzes the conventional (based on the analysis of incomplete phase diagrams) and alternative (based on the analysis of phase diagrams supplemented by a boundary line reflecting the polymer swelling degree dependence on temperature) physicochemical concepts of the thermally induced phase separation (TIPS) method used to prepare UHMWPE membranes. It also considers the main ways to control the structure and properties of UHMWPE membranes obtained by TIPS and the original variations of this method. This review discusses the current challenges in UHMWPE membrane formation, such as the preparation of a homogeneous solution and membrane shrinkage. Finally, the article speculates about the modification and application of UHMWPE membranes and further development prospects. Thus, this paper summarizes the achievements in all aspects of UHMWPE membrane studies.

## 1. Introduction

Membrane technologies hold an important place in modern science and engineering. Separation processes are used for water [1] and wastewater treatment [2], including seawater desalination [3]; filtration of liquid food products, such as vegetable oils [4], juices [5] and alcohol [6]; organic solvent filtration [7]; and gas separation [8], including CO_2_ absorption [9]. Membranes are also used as separators in Li-ion batteries [10,11]. Although a wide variety of materials such as metal oxides [12], silica [13], zeolites [14], metal-organic frameworks [15] and carbon [16] are sometimes used to prepare membranes, polymers remain indisputable leaders among membrane materials [17].

The most common polymers used for membrane preparation are polyvinylidene fluoride [18,19,20,21], polypropylene [22,23], polysulfone [1,21,24], polyethersulfone [25], polyamides [26,27], cellulose acetate [28], nafion [29] and polyethylenes (PE) [22,30].

The latter, in general, being quite cheap, has good thermostability at temperatures up to 100 °C and excellent chemical resistance to acids, alkalis and polar solvents. It is also non-toxic and thus can be used in processes related to food and drinking water. Moreover, PE is a great electrical insulator, which is very important in terms of the separation of anode and cathode in batteries. However, due to its hydrophobic nature, the use of unmodified PE is limited in processes involving polar liquids.

One of the polymers that has attracted increasing attention lately is ultra-high molecular weight polyethylene (UHMWPE). This polymer has better properties than common low-density and high-density PE. In particular, it has much higher wear resistance, mechanical strength, impact strength, and flexural modulus but retains the aforementioned general properties of PE. However, the viscosity of UHMWPE melts and solutions is very high due to its high molecular weight (MW), which makes it difficult to process. Notably, although PE is classified by the ASTM as UHMWPE if its viscosity-average MW is higher than 3 × 10^6^ g/mol [31,32], any PE with MW higher than 10^6^ g/mol is often called UHMWPE in the literature [33,34,35].

Although methods of producing PE with MW higher than 10^6^ g/mol were proposed as early as 1962 [36], this material was of little use and it was not until the late 1990s that it began to be widely applied in arthroplasty. At that time, approximately a million UHMWPE joint replacements were made annually [37]. In addition, UHMWPE has been increasingly applied in other industries, such as the preparation of composite fabrics [34,38,39,40]; UHMWPE fibers have been used to reinforce a variety of composites [41,42].

In recent years, UHMWPE has also attracted researchers’ attention as a potential material for the preparation of porous materials: X-ray shielding composites [34,39], aerogels [43,44,45], and lithium-ion batteries separators [46,47,48,49,50,51,52]. An analysis of the literature published over the past 30 years shows that the problem of obtaining membranes from UHMWPE was first mentioned in 1994, but interest in this problem significantly increased only ~20 years later (Figure 1).

In the last few years, excellent reviews of the progress in preparation, modification and application of the membranes from polypropylene [23], polyvinylidene fluoride [20] and other polymers have been published [1]. However, none of these reviews to date have focused on UHMWPE membranes. The present mini-review of this newly emerged and rapidly growing area aims to fill this gap.

## 2. Methods of Preparation of UHMWPE Membranes

The methods used to prepare membranes from semicrystalline (SC) polymers, including UHMWPE, can be divided into two main groups: solvent-free and solvent-based methods.

The first group of methods includes sintering powders [53], track-etching [54], and one- or two-axis stretching [55,56]. All of these, except track-etching, have been applied to prepare UHMWPE membranes. However, UHMWPE was used in [57,58] to form a superhydrophobic coating on the surface of a polyethylene terephthalate membrane prepared by track-etching. If the term “membrane” is used in a broader sense, bearing in mind that nonporous films are applied in pervaporation processes [59], any technique of monolithic film formation, such as extrusion, hot pressing, calendering, injection molding, and others, can be classified as a solvent-free membrane formation method.

The methods in the second group include immersion precipitation of nonsolvent-induced phase separation (NIPS) [60,61,62,63], thermally induced phase separation (TIPS) [64,65] and their modifications, such as nonsolvent thermally induced phase separation [66,67], vapor [68,69], and evaporation-induced phase separation [70,71], polymerization-induced phase separation [72,73], as well as combined methods that require stretching of the membranes formed by the above methods [74].

Importantly, most of the listed methods are inappropriate for UHMWPE membrane formation. For example, the immersion precipitation method requires the preparation of a homogeneous solution of the polymer in a solvent at temperatures close to room temperature, which is impossible for UHMWPE, as there is no solvent with such high affinity for PE.

Thus, the vast majority of papers devoted to UHMWPE membrane formation through solutions employ the TIPS method or its modifications.

### 2.1. Solvent-Free UHMWPE Membrane Formation Methods

#### 2.1.1. Powder Sintering

The powder sintering method is based on the partial or complete melting of powder particles placed into the desired mold. In this process, the macromolecules on the surface of the molten polymer powders intertwine [75] and the sample becomes structurally integral after cooling. Obviously, this method is unsuitable for fine control of the structure of the obtained capillary-porous bodies, as the pore size is directly dependent on the size of the particles used (note that, generally, nearly spherical particles with unimodal size distribution are preferably used in powder sintering).

In spite of this drawback, the proportion of papers focused on powder sintering as a method of UHMWPE membrane formation [76,77,78,79,80,81,82,83] among all the papers devoted to the problem of UHMWPE membrane preparation is much greater than that for other polymers, such as polypropylene [84] and polyvinylidene fluoride [85,86]. Undoubtedly, powder sintering for the preparation of membranes has advantages because it does not require complex equipment or organic solvents. However, in our opinion, the wide use of powder sintering for membrane preparation from UHMWPE is largely the result of insurmountable difficulties in UHMWPE membrane formation by other conventional methods.

Figure 2 shows examples of a typical structure of membranes prepared by the sinering of UHWMPE powders [76,80,81]. The membrane is an aggregate of partially connected spherical or potato-like particles. The size of the pores (voids between the particles) depends on the particle size distribution of the powder used. Since the compactization process during sintering is not ideal, the membrane structure can contain macrovoids up to hundreds of micrometers in diameter. Figure 2b,c show defects on the surface of some of the particles on the cleavage of the samples. The presence of such defects suggests that only thin surface layers of the powder particles participate in the formation of continuity of the samples, while the material in the internal layers of the particles remains unmixed.

The structure and transport properties of the obtained membranes are controlled by:Changing the sintering time [77,78] that determines the duration of interdiffusion of the macromolecules in the molten surfaces of the particles;Changing the polymer MW [78], which determines the rheological properties of the polymer melt;Raising the external pressure to 69 MPa [80] and the temperature of sintering from 135 °C [77] to 250 °C [87].

Original methods of UHMWPE powder sintering were proposed in [77,79,81,83,88]. For example, in [77], powder sintering was performed in an inert liquid (80 vol% of glycerol) and in the presence of 2–4 vol% of plasticizer (m-xylene or paraffin oil). In [79,83,88], a dry powder was mixed with NaCl crystals in ratios ranging from 1:1 to 1:9 before sintering. Then, the salt was removed from the membrane (obtained by hot pressing) by sonication in a water bath or washing at 250 bar. In [81], sintering was performed during the rotation of the vessel with the powder.

The membranes obtained by powder sintering from UHMWPE usually have quite high mechanical strength (24–36 MPa) and a high pure water flux (from ~12 [78] to ~2500 [80] l/m^2^ min atm) but relatively low overall porosity compared to microfiltration membranes obtained from other polymers by TIPS and immersion precipitation. However, the UHMWPE membranes prepared by powder sintering also have some drawbacks. In particular, most of these membranes are several millimeters thick, while common microfiltration membranes are several hundred micrometers thick. In addition, as shown in [80,82], the membranes prepared by sintering have quite a wide pore size distribution. For example, in the membrane with an average pore size of 2.16 µm, the maximum through-pore size was 13.41 µm. Thus, the separation performance of the discussed membranes in a typical microfiltration process is rather poor.

Interestingly, the problem of UHMWPE powder sintering was also discussed in [87,89,90,91,92,93,94,95], but these studies did not aim to prepare membranes. Rather, in these papers monolithic films or composites were prepared and fundamental aspects of the UHMWPE sintering process were addressed. It was shown [87,89] that the continuity of the samples after sintering was not only the result of macromolecule reptation at the boundary between the particles but also of the so-called melting explosion process. The latter process leads to a fast sideway motion of the macromolecules when the polymer crystallites melt, thus ensuring entanglement between the macromolecules in different powder particles.

#### 2.1.2. Stretching

Stretching of monolithic films is widely used to prepare porous structures from a variety of SC polymers [55,56]. The behavior of SC polymers in the process of stretching has been the subject of a large number of fundamental studies published recently [96,97,98,99,100,101,102,103,104,105,106,107]. These papers present detailed studies of the mechanism of craze forming and processes that happen in a nano-scale sample (such as lamellae fragmentation [104,105,106], lamellae slipping [100,106], recrystallization [108], microbuckling [97], etc.). The features of UHMWPE stretching are discussed in [109,110,111,112,113,114,115,116].

Monolithic UHMWPE films used as the starting material in membrane formation via stretching are obtained by hot pressing of the powders at a temperature lower than or higher than the polymer melting temperature [91,117,118,119,120], calendaring [121], skiving off the block [122], etc.

To date, only a small number of publications have reported UHMWPE membrane formation by stretching [112,123,124]. Many more papers have focused on UHMWPE membrane formation by a combination of TIPS and stretching.

A typical structure of the membranes prepared by stretching is shown in Figure 3 [123]. The structure of the membranes is usually controlled by changing the stretching temperature, draw ratio and method (one- and two-axis stretching).

### 2.2. Solvent-Based Methods of UHMWPE Membrane Formation

Among the numerous methods of membrane formation from SC polymers through solutions, only the TIPS method is suitable for UHMWPE membrane preparation. This method was proposed in 1981 [125] and nowadays is one of the most common methods of membrane preparation from SC polymers. A typical TIPS process starts with the preparation of a homogeneous mixture of a polymer and a diluent (usually a low MW liquid) at an elevated temperature. Then, a polymer solution is shaped into a desired form (such as a flat sheet or hollow fiber) with a casting knife, extruder, etc. Later, the thermal energy is removed from the sample either on air or in an inert liquid medium. At low temperatures, the solution becomes thermodynamically unstable and thus cooling of the mixture induces phase separation and polymer crystallization to produce a porous structure. Then, the solvent is removed from the pores of the formed membrane by extraction or annealing in vacuo.

#### 2.2.1. Conventional Physico-Chemical Basis of the TIPS Method

The first attempt to formalize the TIPS process of an SC polymer mixture with a solvent and to connect the obtaining structures with the thermal behavior of the mixtures based on temperature-composition phase diagrams was made in 1985 by W.C. Hiatt et al. [126]. However, a series of papers entitled “Microporous membrane formation” published in the early 1990s by D.R. Lloyd et al. [64,65,127,128,129,130,131] received much greater recognition. In this series of papers, the authors plotted experimental phase diagrams, discussed the mechanisms of membrane formation and studied the effect of membrane formation process parameters (such as cooling rate, solvent and polymer nature, concentration and MW of the latter, presence of a nucleating agent, etc.) on the structure of the obtained membranes. The understanding of structure formation in mixtures of SC polymers with low MW liquids during their TIPS, proposed by Lloyd et al., has undergone only some minor changes and has survived to this day in an almost unchanged form. For example, this understanding has been briefly summarized in recent reviews [20,22,132,133,134].

It follows from the cited papers [20,22,64,65,132,133,134] that phase separation of homogeneous SC polymer mixtures with low MW liquids can be realized either by solid–liquid phase separation [64] (in the case of mixtures of SC polymers with so-called good solvents) or by liquid–liquid separation [65] (in the case of mixtures of SC polymers with so-called poor solvents). The events taking place in the process of cooling homogeneous mixtures were discussed using the schemes of phase diagrams shown in Figure 4I,II [20]. However, a literature analysis shows that obtaining a complete picture requires considering one more phase diagram type. The diagrams of the third type (Figure 4III) reflect the thermodynamic behavior of the SC polymer mixtures with a low MW crystalline substance with a melting temperature comparable to that of the polymer.

The phase diagram of the SC polymer mixture with a poor solvent (Figure 4I) contains a binodal curve (blue) and a polymer crystallization line (orange) intersecting at a monotectic point (red).

Cooling of the blend according to path (A) results in crossing binodal and spinodal curves and leads to liquid–liquid phase separation via spinodal decomposition. The compositions of the coexisting phases are determined by the intersection of the tie lines with the left and right branches of the binodal curve. Then, after crossing the orange line, the polymer crystallizes. In addition to the existing liquid phases, the crystallization process results in the formation of one more phase—polymer crystals that do not contain liquid. However, the question of what happens to such a three-phase system during subsequent cooling has not received any attention [20,22,64]. In any case, the bicontinuous structure shown in Figure 4a is formed after cooling to room temperature and solvent removal.

Cooling of the blend according to (B) path also leads to separation of the mixture into two liquids and subsequent polymer crystallization. The only difference is the nucleation and growth mechanism that drives liquid–liquid phase separation. (Note that cooling of the mixture with the composition to the left of the upper critical solution temperature (critical point in Figure 4I) also results in liquid–liquid phase separation by the nucleation and growth mechanism. However, in this case, the phase depleted by the polymer acts as the dispersion medium, while the polymer-enriched phase comprises the dispersed phase. In this case, the result of TIPS is a suspension of the polymeric powder instead of a porous membrane; thus, this path is usually not paid attention to). The TIPS process is followed by the formation of a cellular structure, as shown in Figure 4b. It should also be noted that the transition between spinodal decomposition and nucleation and the growth mechanism is not sharp. Moreover, the preferred mechanism of liquid–liquid phase separation depends not only on the mixture composition but also on the cooling rate [65,135]. Thus, the transition from the bicontinuous structure to the cellular one happens gradually upon the decrease in the cooling rate and increase in the polymer concentration. In the limiting case, a decrease in the cooling rate can lead to decomposition of the mixture into two macrophases separated by a single horizontal interface.

Cooling of the mixture according to path (C) results in solid–liquid phase separation after crossing the polymer crystallization line. Thus, a crystalline phase that does not contain liquid emerges and the composition of the liquid that surrounds the crystals formed moves along the crystallization line until it reaches the monotectic point where it undergoes liquid–liquid phase separation [22,65]. Some works [20,22,127] state that the separation of the crystalline phase goes through the nucleation and growth mechanisms. The transformations that occur in the system during subsequent cooling are not discussed. The structure formed can be classified as a combination of spherulites impinged by every surface, as shown in Figure 4c.

The scheme of the type II phase diagram reflects the thermodynamic behavior of the SC polymer mixture with a good solvent. Such diagrams also contain a polymer crystallization line and a binodal curve, but the latter is fully located under the polymer crystallization line. Interestingly, in Lloyd et al.’s original papers [64] and in the experimental phase diagrams published later [136,137,138,139,140], there is no binodal curve on the diagrams of such a type. Although the principal topology of such diagrams containing the binodal curve was discussed as early as 1971 [141], from the thermodynamics point of view, such topology is only possible if one of the boundary curves is not an equilibrium one. Meanwhile, the mechanisms of structure formation in such mixtures to this date are usually discussed [] using a phase diagram containing a binodal, as shown in Figure 4II.

Cooling of the mixture by path (D) leads to the formation of particle-like structures connected by leaves, tied fibrils, fuzzy spheres and leafy spherulites (examples of such structures are shown in Figure 4d,e). As far as we are aware, the mechanism of the formation of such structures has never been discussed in detail. This is probably due to the fact that membranes with such structures have a much lower mechanical strength than those with structures shown in Figure 4a,b,f,g,h and thus are not very promising from a practical point of view [142,143,144]. For example, in reviews [133,134], this problem did not receive any attention. The authors showed only the micrographs of structures formed after cooling by this path [20,132]. In other papers [22,64], where this problem was discussed to some extent, the process was considered using phase diagrams that contained only the polymer crystallization line (without the binodal curve). According to the scenario proposed in these papers, crossing the crystallization line, regardless of the initial polymer concentration, resulted in the formation of polymer crystallite nuclei due to heterogeneous nucleation and their growth into spherulites due to secondary nucleation. However, the causes of the formation of different structures, such as those presented in Figure 4d,e, were not discussed. The mechanism of formation of interspherulitic pores that are clearly visible in [145,146], for example, was not presented. In addition, as we showed in [147], the application of the lever rule to the analysis of the phase diagram in Figure 4II yielded an incorrect prediction that the crystallinity degree of the polymer below the crystallization line must approach 100%.

Figure 4III presents the scheme of the phase diagram for the SC polymer blends with a low MW crystalline substance similar to the one presented in the experimental phase diagrams for such systems in [130,148,149,150,151,152]. Phase diagrams with such topology are commonly considered eutectic phase diagrams containing two liquidus lines of the components and a horizontal solidus line intersecting at the eutectic point.

Cooling of the mixture by path (F) results in crystallization of the low MW component after crossing the diluent crystallization line, while the composition of the liquid surrounding the crystals formed moves along the diluent crystallization line. When this composition reaches the eutectic point, both of the liquid components rapidly crystallize. Thus, cooling of this blend leads to the formation of a dispersion of the diluent crystals that grew from the liquid phase in the thin dispersion of the remaining components. Removal of the diluent from the samples results in the formation of structures similar to those shown in Figure 4f. The shape of the large pores depends on the preferred shape of the diluent crystals. Cooling the mixtures with the composition close to the eutectic point composition (path (G)) leads to the fast formation of a thin dispersion as a result of simultaneous crystallization of the components. An example of the capillary-porous body structure formed from such a mixture is shown in Figure 4g. Finally, cooling the mixture by path (H) on the phase diagram leads to crystallization of the polymer after crossing the orange line, while the composition of the surrounding liquid moves along the polymer crystallization line. After crossing the solidus line, both components rapidly crystallize, as in the above cases. The described scenario suggests that the final structure is a dispersion of spherulites in a thin dispersion of both components. However, the real structure shown in Figure 4h is different. Another view of the TIPS process in an SC polymer mixture with an LM crystalline substance is proposed in Ref. [153]. These papers state that the crystallization of the components proceeds independently and only the onset crystallization temperature of each of the components depends on the initial blend composition. A significant drawback of the discussed points of view is the following contradiction. According to the IUPAC goldbook [154], solidus is a line on a binary phase diagram (or a surface on a ternary phase diagram) that indicates the temperature at which a system becomes completely solid on cooling or at which melting begins on heating under equilibrium conditions. This definition suggests that, below the solidus line, only crystals of both components may exist. Meanwhile, the crystallinity degree of real samples of SC polymers is well known to be far from 100%. Thus, the proposed understanding, based on the assumption that the phase diagram for the SC polymer mixture with a crystalline low MW substance is eutectic, cannot be correct.

An analysis of the above allows the conclusion that the phase diagrams with the topology shown in Figure 4 cannot be used to describe the TIPS mechanisms in different mixtures or to predict the structure of the membranes formed from them.

#### 2.2.2. Alternative Physico-Chemical Basis of the TIPS Method

An alternative view of the process of structure formation during TIPS was proposed and presented by us in Refs. [147,155,156,157,158,159]. We were able to develop this point of view by adding one more boundary curve to the phase diagrams shown in Figure 4. The curve reflects the dependence of the polymer swelling degree on temperature or is the same as the dependence of the L MW component solubility in the SC polymer amorphous regions on temperature (the red curve in Figure 5). Although this curve was for the first time plotted on a phase diagram by R.B. Richards in 1946 [160], its significance for describing the thermal behavior of SC polymer mixtures with low MW substances has not been given due attention. As far as we are aware, it is only in our work that this curve has been experimentally constructed since then [147,155,156,157,158,159,161,162,163,164]. However, it was also plotted on the schematic phase diagrams in a recent review [165] and was shown as a hypothetical extrapolation of the binodal curve in Refs. [166,167,168,169,170]. The absence of this curve on the published phase diagrams, in our opinion, is explained by the fact that the common methods used to plot phase diagrams, such as differential scanning calorimetry (DSC) and the cloud point method, are not suitable for finding its position. However, the published data on the temperature dependence of the swelling degree of a polymer [171,172], obviously, can be used to reconstruct the position of this curve on a temperature-composition phase diagram.

Importantly, the presence of this curve drastically changes the thermodynamic meaning of the phase diagrams. The method of phase diagram construction, the physical meaning of the curves on the phase diagrams of different types and the state of the mixtures in different areas were discussed in detail in [147,155,156,157,158,159,161,162,163,164]. Figure 5 shows schemes of the phase diagrams plotted based on the data from [147,156,159] for real mixtures of SC polymers with low MW substances. The physical meaning of the curves is shown by blue, red and orange text. The black text indicates the process that takes place in the respective regions of the phase diagram when the mixture enters this region upon cooling. An analysis of the phase diagrams with the topology shown in Figure 5 allowed us to develop the following understanding of the structure formation mechanisms during the TIPS of different mixtures.

Cooling of the mixtures by paths (A) and (B) results in liquid–liquid phase separation via spinodal decomposition (A) or nucleation and growth (B) after crossing the binodal. The emerged polymer-rich and polymer-lean phases form an emulsion of the latter in the former. The compositions of the coexisting phases change along the left and the right branches of the binodal until the polymer-rich phase reaches the composition where the binodal curve and polymer crystallization line intersect (the red point of Figure 5I). At this moment, the polymer starts to crystallize. Notably, since the crystallization starts in the polymer-rich phase with the composition corresponding to the red point, regardless of the initial composition, the polymer crystallization line fragment below the binodal must be strictly horizontal from the thermodynamic point of view.

An extract from the IUPAC goldbook [173] states that a “gel is a non-fluid colloidal network or polymer network that is expanded throughout its whole volume by a fluid that can contain a polymer network formed through the physical aggregation of polymer chains, caused by hydrogen bonds, crystallization, helix formation, complexation, etc., that results in regions of local order acting as the network junction points. The resulting swollen network may be termed a thermoreversible gel if the regions of local order are thermally reversible”. Bearing that in mind, one can assert that the crystallization of an SC polymer is a process of gel formation and gel in this case is a swollen SC polymer network. In addition, the swelling degree of an SC polymer in a solvent is usually known to decrease with a temperature reduction. Therefore, when cooling continues, one more process is expected to occur—deswelling of the polymer that results in separation of the liquid that remains dissolved in the polymer amorphous regions.

Thus, contrary to conventional opinion, crossing the crystallization line leads not only to the crystallization of the polymer that fixes the structure formed. Rather, it results in two continuous processes: gel formation (formation of a three-dimensional network of crystallites connected by tie chains and entangled loops) and solvent separation from the amorphous regions of the gel. The latter process is also accompanied by the relaxation of the polymeric matrix. The structure of the formed capillary-porous body depends on the relaxation rate. If the relaxation rate is sufficiently high, the amorphous regions shrink, the polymer deswells and the separated solvent increases the volume of the polymer-lean phase droplets formed during the liquid–liquid phase separation. If the relaxation rate is not high enough, polymer deswelling results in the formation of small droplets of the solvent inside the amorphous regions of the polymer. The foregoing explains the formation of small pores in the walls of the cellular or bicontinuous structure observed in [147,174,175,176].

Cooling of the mixture by path (C) leads, after crossing the inclined polymer crystallization line, to the formation of the gel throughout the sample volume due to nucleation, growth and impingement of the spherulites. The gel formed remains macroscopically uniform as long as the system stays in the region between the orange and the red curves. However, its cooling below the swelling curve results in solvent separation from the amorphous regions of the gel, usually in the form of small droplets inside the spherulites. The resulting structures can be classified as spherulites impinged along all surfaces, with small pores inside them. A more detailed discussion of the realized processes can be found in [147,158].

The phase diagram of type II shown in Figure 5II can be used to conclude that, regardless of the initial composition (for example, path (D) and (E)), cooling results in events similar to those discussed for path (C). Namely, crossing the polymer crystallization line is followed by gel formation and crossing the red curve leads to solvent separation from the gel as a result of its deswelling. Let us discuss these processes in more detail. Cooling of the blend of any composition leads to spontaneous nucleation and growth of the spherulites or other polycrystalline structures directly in the volume of the homogeneous solution. The macromolecules seem to be “pulled” by the growing crystallites, while the solvent, as stated in [138,177,178], is rejected by the growing crystallites. It should be noted that the thermodynamic rule is that the polymer crystallites must not contain liquid, but the latter can still be located inside spherulites with amorphous regions. Thus, a certain amount of the solvent remains solvated by the amorphous regions of the macromolecules between the lamellar crystallites. After crossing the red line of the phase diagrams shown in Figure 5II, this solvent separates into its own phase, either forming small droplets inside the amorphous regions of the polymer or joining the liquid between the spherulites if the relaxation process is sufficiently fast (also leading to shrinkage of the spherulites). Note that such small intraspherulitic pores were detected in Refs. [179,180,181,182,183].

The polymer concentration affected only the size and quantity of the formed spherulites. If the polymer concentration is low, the spherulites stop growing before they impinge with each other and a suspension of porous particles in the solvent is formed. A higher polymer concentration increases the probability of impingement of the growing spherulites and at some critical concentration, a continuous structure is formed that comprises porous spherulites impinged along some surfaces with the solvent in voids between them (Figure 5d). This critical concentration depends on several factors, such as the polymer MW, thermodynamic affinity between the mixture components, cooling rate, etc. and usually is about 10–20 vol%. There is also another critical polymer concentration in the mixture corresponds to the impingement of the spherulites along all the surfaces. This concentration differentiates between the structures shown in Figure 5d,e. A more detailed discussion of the phase diagrams of this type and structure formation can be found in our previous papers [147,158,159]. It should be noted that although spherulitic crystallization is the most common process, crystallization can also result in the formation of leaves [184], nodules [185], etc. However, this does not change the foregoing discussion.

As stated above, the type III phase diagram is not a full analogue of the eutectic phase diagram for a mixture of two low MW substances. Figure 5III shows the phase diagram for an SC polymer mixture with a low MW crystalline substance implemented with a swelling curve. Cooling of the mixture by path (F) leads to crystallization of the low MW substance from its solution in the polymer melt. The composition of this solution during subsequent cooling moves along the blue line, while the mass fraction of the low MW substance crystals increases. After crossing the orange line, the polymer crystallizes and forms gel in the liquid surrounding the low MW substance crystals formed. Subsequent cooling results in the separation of the low MW substance from the amorphous regions of the polymer, usually in the form of small crystals. The presence of such crystals in the cooled samples is confirmed by the appearance of endothermic peaks on the second heating thermograms in DSC experiments, which reflect the melting of such crystals at temperatures lower than that of the pure low MW substance [155,156]. As a result, after the diluent is removed, the structures formed comprise a polymer matrix with large pores (replicating the shape of the low MW substance crystals) and small pores, both uniformly distributed (Figure 5e). In the case of mixture cooling along the G path, the crystallization of the polymer and of the low MW substance begins simultaneously when the crystallization line (at the red point in Figure 5III) is reached, and the mixture structure depends on the ratio of the crystallization rates of the components. In this case, as well as in the case of mixture cooling along path (F) below the polymer crystallization line, the gel undergoes microphase decomposition in accordance with the swelling line. The resulting structure is shown in Figure 5g. The events that occur upon cooling of the mixture corresponding to path (H) differ from the events that occur when mixtures are cooled along paths C and E only in that below the swelling line; the low MW substance is separated from amorphous regions in the form of small crystals rather than small liquid droplets. The removal of the low MW component from the mixture leaves a structure, as shown in Figure 5h.

An analysis presented in this section indicates that the alternative ideas about the physicochemical basis of the TIPS method, based on the analysis of phase diagrams with a new topology, make it possible, in contrast to conventional ones, to correctly describe the thermal behavior of mixtures upon cooling and to explain the mechanism of the formation of porous structures. We hope the reader will find the new concepts [147,158,159] summarized in this section useful when reviewing the literature and developing new technologies for obtaining membranes using the TIPS method.

#### 2.2.3. Problem of Dope Solution Preparation

As noted above, researchers developing new technologies for the preparation of UHMWPE membranes via TIPS encounter difficulties already at the stage of solvent selection. There are several reasons for this. First, there are no solvents capable of dissolving pure polyolefins (including PE) at room temperature. Aromatic (toluene, xylenes and other alkylbenzenes) and aliphatic (n-alkanes, branched and cyclic alkanes) hydrocarbons, which are considered to be the best solvents for PE, as well as some chlorine derivatives (for example, 1,3,5-trichlorobenzene) form homogeneous mixtures with relatively low MW polyethylenes (LDPE, HDPE) at temperatures of ~70 to 100 °C, depending on the polymer concentration and nature. Second, the thermodynamic affinity of low MW liquids for PE is known to decrease with an increase in the polymer MW [186]. Therefore, solvents widely used for membrane preparation from common polyolefins, but having a somewhat worse thermodynamic affinity for the polymer (soybean oil [174], DOP [187], etc. [188]), cannot form molecular mixtures with UHMWPE at technologically reasonable temperatures. Third, due to the high MW of the discussed polymer, the viscosity of its solutions in low MW liquids is significantly higher than that of the solutions of common polyethylenes with the same concentration. Thus, it is rather difficult (even for a solvent with good thermodynamic affinity for PE) to obtain homogeneous mixtures of components within a reasonable time.

Our analysis of the literature concerning the problem under discussion made it possible to summarize the data on the conditions for preparing UHMWPE solutions in various solvents in Table 1.

An analysis of the data presented in Table 1 allows us to conclude the following.

Almost half of the published works report using various antioxidants that prevent degradation of polymer macromolecules for successful preparation of homogeneous UHMWPE solutions.Although the MW of the polymers used to prepare the solutions covers a wide range of values (including the values that are beyond the range of values corresponding to the UHMWPE definition) from 0.5 × 10^6^ to 9 × 10^6^ g/mol, UHMWPE with MW ~3 × 10^6^ g/mol is the most widely used.In most cases, solutions are produced at temperatures of 160–250 °C, which is significantly higher than the melting point of the pure polymer (135–145 °C) under conditions of high shear (twin screw extruders, batch mixers, rheometers). Often, a preliminary step of thorough mixing of the polymer with the diluent at slightly elevated or room temperature is added.In more than 60% of the cited works, the authors use liquid paraffin (LP), mineral, white and paraffin oils as the solvent for UHMWPE. Interestingly, these are essentially different names for the same object—a liquid mixture of linear and/or branched saturated hydrocarbons. Moreover, as a rule, the fractional composition of the solvent is not disclosed. Only a few works [33,50,198,199,200,201,202] specify the MW range of the components in LP, usually ranging from 150 to 300 g/mol. In [198], the authors used a mixture of hydrocarbons with an average molecular weight of 500 g/mol, which should probably be classified not as LP but as paraffin wax.Only a few studies have employed other substances as solvents for UHMWPE. In particular, it was proposed to use decalin, xylene, naphthenic oil, diphenyl ether, etc. A slightly larger variety of potential solvents have been proposed in patents [204,206,217], for example, dioctyl phthalate, dibutyl phthalate, aromatic oils, etc. However, LP is considered the best choice.

Thus, in our opinion, the problem of selecting the solvent and conditions for the preparation of UHMWPE solutions is still far from completely solved.

#### 2.2.4. Investigation of Thermal Behavior of UHMWPE Mixtures with Various Solvents

Taking into account that, as Section 2.2.1 and Section 2.2.2 show, structure formation during the TIPS process can be driven by different mechanisms depending on the qualitative and quantitative composition of the initial mixture. Traditionally, the first stage in the development of new technologies for the preparation of membranes via TIPS is usually the study of the thermal behavior of the chosen mixture. The main result of such a study is the temperature–composition phase diagram. This practice, however, is much less common when preparing UHMWPE membranes [33,191,195,198,199,200,202,203,209,230], probably because it is difficult to construct phase diagrams for mixtures containing such a high MW polymer.

Only two papers [198,230] reported the phase diagram for a mixture of UHMWPE and a low MW substance plotted over the entire range of compositions from 0 to 100%. These phase diagrams are shown in Figure 6.

According to the authors of [230], both phase diagrams shown in Figure 6a have a topology of type I, i.e., they contain a liquid equilibrium binodal and a polymer crystallization line located below the binodal. It can be seen that the mass fraction of the polymer at the upper critical solution temperature (UCST) in the phase diagram of UHMWPE with decalin is ca. 40%, which is extremely atypical of polymer mixtures with a low MW substance. Usually, the composition corresponding to the UCST in such mixtures is strongly shifted to the region of compositions enriched by a low MW substance. This allows us to assume that the phase diagram for the UHMWPE mixture with decalin actually belongs to type (II), and all the points (squares in Figure 6a), taking into account the experimental error, correspond to the polymer crystallization line. This point of view is also confirmed by scanning electron microscopy (SEM) images of the structure of capillary-porous bodies published in the same work [230], obtained by cooling mixtures containing 40 wt% polymer (Figure 7). One can see in this figure (Figure 7a) that the porous body obtained from a mixture of UHMWPE with diphenyl ether has a cellular structure resulting from liquid–liquid phase separation followed by polymer crystallization. This type of structure was to be expected, taking into account the topology shown in Figure 6a (see the circles in Figure 6a) and the conclusions made in Section 2.2.1 and Section 2.2.2. At the same time, Figure 7b shows that the porous body obtained from a mixture of UHMWPE with decalin (see the squares in Figure 6a) has a spherulitic structure, which, as shown in Section 2.2.1 and Section 2.2.2, must be the result of so-called solid–liquid decomposition. However, the authors of [39], who did not plot the phase diagram of the respective system, also stated that the UHMWPE–decalin mixture underwent solid–liquid phase separation.

The phase diagram for the mixture of UHMWPE with LP, shown in Figure 6b, also belongs to type II. Moreover, to make sure no liquid decomposition occurs in the system, the authors obtained a diagram using two methods: DSC, which fixes only the process of polymer crystallization, and light scattering, which can observe turbidity resulting either from liquid decomposition or from polymer crystallization. The fact that the temperature values obtained by different methods are essentially the same led the authors of [198] to conclude that the only process responsible for the structure formation upon cooling of UHMWPE mixtures with LP is polymer crystallization. The same opinion is shared by the authors of [212]. In the cited paper, the phase diagram of the UHMWPE mixture with LP was plotted in the polymer composition range of 5 to 35 wt%.

However, an analysis of other papers [33,34,202] indicates that even such a basic question regarding the type of phase diagram of a UHMWPE mixture with LP is still controversial. In particular, in [202], despite the fact that, according to the topology, the plotted diagram belongs to type II, the authors speak of UCST-type phase behavior in the system.

The potential possibility of liquid decomposition in UHMWPE mixtures with liquid paraffin has also been reported in Refs. [33,34]. In [33], a thorough analysis of the data on the rheology of UHMWPE mixtures with LP in various cooling and heating modes showed that in the range of polymer concentration values from 25 to 60 wt%, polymer liquid decomposition is fixed at temperatures of 2.5–6.5 °C above the polymer crystallization temperature at a cooling of 0.2 °C/min.

It is usually assumed that liquid decomposition is much faster than polymer crystallization [169,235,236,237]; therefore, if the binodal lies above the crystallization line on the dynamic phase diagram, the structure of the capillary-porous body obtained upon cooling is always the result of liquid decomposition followed by crystallization. In the case of UHMWPE, the authors of the cited work believe that, on the contrary, the process of liquid decomposition is more dependent on kinetics. Thus, despite the fact that the phase diagram belongs to type I, it is possible to obtain membranes with a structure characteristic of liquid decomposition only by long-term annealing of samples in the region between the liquid equilibrium binodal and the crystallization line. Note that for common polymers, the situation is completely different: if the phase diagram belongs to type I, it is only very fast cooling that can yield non-cellular/non-bicontinuous structures for polymer-depleted mixtures. Figure 8 shows a phase diagram from the paper in question and an SEM image of the structure of the membrane obtained according to the procedure discussed.

The phase diagram in Figure 8a turned out to be very useful in practical terms, as it made it possible to predict the possibility of liquid decomposition in a mixture. However, it raised some questions from a thermodynamic point of view. If we interpret the black squares as points corresponding to the binodal curve, the composition corresponding to the UCST must be at least 57 wt% of the polymer, which causes doubts about the correctness of their position. At the same time, the data on the rheology and structure of the capillary-porous bodies obtained in various cooling modes (including those shown in Figure 8b) unambiguously indicate that the liquid–liquid decomposition in the mixture took place before the polymer crystallization. Thus, it can be assumed that the black squares in Figure 8a correspond to the temperatures at which the method used at the sensitivity limit detects the liquid decomposition of the mixture but not to the equilibrium position of the binodal, which lies somewhat higher, taking into account the presumed position of the UCST in the low-concentration region.

#### 2.2.5. Factors Controlling the Structure and Properties of the UHMWPE Membranes Prepared via TIPS

One of the main tasks of materials science is to obtain materials with desired properties (in the case of membranes, with the desired structure, transport (performance, rejection, etc.) and mechanical (strength, elongation at break, etc.) and other properties). To control these characteristics, different parameters, such as polymer concentration in the dope solution, MW of the polymer, solvent quality, cooling conditions, etc., are varied during membrane preparation.

##### Polymer Concentration in the Dope Solution

Typical values of UHMWPE concentrations in the dope solution for preparing membranes via TIPS lie in the range from 5 to 50% of the mass. Importantly, this range is wider than the typical concentration range of dope solutions used to prepare membranes via TIPS from common polymers, such as PP and PVDF (typically 15–30 wt%). These values are considered optimal since the porosity of the membranes prepared via TIPS, which affects both the transport (the higher the porosity, the better) and mechanical (the lower the porosity, the better) properties, is usually equal to the volume fraction of the solvent in the initial mixture (or less if shrinkage occurs at any stage of membrane preparation). The polymer concentration in the dope solution of ca. 15–30 wt% ensures the formation of membranes with the best combination of transport and mechanical properties.

The extension of this range to lower polymer concentrations is associated with two factors. The first of them is purely technological: since the solution viscosity grows with an increase in the polymer MW, solutions with a polymer concentration of ~15–30% turn out to be too viscous and difficult to process [205]. It is obvious that the viscosity can be reduced by lowering the polymer concentration in the initial solution. As noted above, the lower limit of polymer concentration in a typical dope solution is determined by the requirements for the mechanical strength of the resulting membranes. Due to the high MW and, consequently, the large specific number of entanglements, UHMWPE provides acceptable values of the mechanical strength of products and structure continuity, even at a fairly low concentration of the initial solution. In addition, due to intense shrinkage, the porosity of the membranes obtained from UHMWPE was significantly lower than the volume fraction of the solvent in the initial mixture. On the one hand, this means that the initial UHMWPE solution must have a lower concentration to achieve the same values of porosity if shrinkage is taken into account. On the other hand, other things being equal, shrinkage makes the matrix area in the sample cleavage surface larger and, consequently, increases the mechanical strength. The combination of these factors makes it possible to use UHMWPE solutions with a polymer concentration of 10% or less to prepare capillary porous bodies.

The extension of this range to a region of more than 30 wt% is used when the membranes are prepared by a combined method, including TIPS and subsequent (or parallel) drawing, which provides an increase in porosity.

Since the problem of high viscosity and processability of UHMWPE solutions is quite acute, a lot of attention in some papers [195,215,218,219,231] has been paid to the study of the rheological properties of initial solutions, including the dependence of these properties on the initial mixture composition.

The effect of the UHMWPE concentration in the initial solution on the structure and properties of membranes was evaluated in Refs. [50,209,230]. It was shown that an increase in the polymer concentration led to higher mechanical strength and lower porosity and reduced the average pore size and flux, which is typical of membranes obtained by the TIPS method from any SC polymer. In some cases [209], however, the transport properties of membranes pass through an extremum as the polymer concentration in the initial solution increases. This is explained by the fact that despite the continuous decrease in theoretical porosity with an increase in the polymer concentration in the mixture, membranes obtained from mixtures of different concentrations can have different shrinkage degrees.

##### MW of the Polymer

As Table 1 shows, the molecular weight of UHMWPE used to prepare membranes by TIPS usually ranges from 10^6^ to 4*10^6^ g/mol. As mentioned above, with an increase in the polymer MW, it becomes increasingly difficult to process solutions of the same concentration. The influence of UHMWPE MW on the rheological properties of solutions and crystallization kinetics was studied quite systematically in Refs. [195,203]. To solve the problem of processability, some researchers have proposed blending UHMWPE with other polymers, such as PE with lower MW [52,109,203,233] and PVDF [208,211,215]. The influence of the polymer MW on the topology of the phase diagram of its mixture with a low MW liquid was studied in detail for PE [238], PP [186], PVDF [239], and other polymers [240]. These papers showed that the position of the polymer crystallization line (Figure 4 and Figure 5) weakly depends on the polymer MW, while the binodal curve (and, consequently, the swelling [148,158] curve) shifts to higher temperatures. Although there is no direct evidence in the literature for such behavior of UHMWPE mixtures with low MW liquids (moreover, the binodal with a pronounced UCST was obtained only in [230]), in our opinion, there are no grounds to believe that the data obtained for PE, as an example, with MW in the range from 10^4^ to 5 × 10^5^ g/mol are not relevant for UHMWPE. Papers [195,201,203] report a slight increase in the temperature of crystallization [52], cloudiness/clearing [203] or phase decomposition [195] with an increase in the UHMWPE MW.

As for the MW effect on the properties of the membranes prepared by TIPS, it was shown in [201] that the increase in the UHMWPE MW from 2.5 × 10^6^ to 4 × 10^6^ g/mol, lowered the water flux of composite membranes by ~30–40% (depending on the polymer concentration in the initial mixture from ~1200 to 800 or from 420 to 230 l/m^2^ h), and increased the BSA retention coefficient (at the same polymer concentrations from 15 to 53 or from 40 to 80%). The authors attributed these facts to a decrease in pore size as a result of the slowing down of the solid-liquid phase decomposition.

##### Solvent Nature

Only a small number of papers have addressed the influence of a solvent on the thermal behavior of its mixtures with UHMWPE and the structure and properties of membranes prepared from these mixtures [195,230]. The results from these papers are in good agreement with the general trends of the influence of solvent thermodynamic quality with respect to the polymer on the topology of phase diagrams, etc. As the quality of the solvent improves, both the binodal curve and the polymer crystallization line shift to lower temperatures and the former is more sensitive to the thermodynamic affinity of the solvent for the polymer. In the limiting case, the binodal curve disappears and the phase diagram topology changes from type I to type II. This is how, in our opinion, the data of Ref. [230] on thermal behavior and the corresponding changes in the structure of membranes obtained using diphenyl ether and decalin should be interpreted. In the case of a UHMWPE mixture with dipehynil ether (a poor solvent), the phase diagram for its mixture with UHMWPE contains a binodal curve, and the membranes obtained from this mixture have a cellular structure. In the case of a UHMWPE mixture with decalin (a good solvent), there is no binodal curve on the phase diagram, and the membranes obtained have a spherulitic structure.

It was shown in [195] that the polybutene oligomer (M ~400 g/mol) has a worse affinity for UHMWPE than LP, but the predominant type of phase decomposition in both systems is solid–liquid separation. It is noted in [39] that although the use of LP as a solvent under certain conditions makes it possible to achieve higher permeability values, the use of decalin also has its advantages (in particular, a lower temperature is needed to prepare a homogeneous initial solution).

##### Cooling Conditions

It is well known that the structure and properties of membranes obtained via TIPS depend significantly on the cooling rate. Early data on the cooling rate effect on the structure of the membranes prepared via TIPS were published as early as the 1990s in D.R. Lloyd’s fundamental works [65,127,131]. According to these papers, as the cooling rate increases, the pores and spherulites become smaller and the cellular structure can be transformed into a bicontinuous one. With regard to membranes made of PP, PVDF and other polymers, the cooling rate has long been and remains one of the main ways to control the pore structure [22,241]. In the case of UHMWPE, diffusion processes in the mixtures proceed slowly due to the high MW of the polymer; therefore, this method is less effective. It was shown in [33,34] that an increase in the annealing time of UHMWPE mixtures with LP at temperatures higher than 110 °C may transform the porous structure of the formed membranes from the type characteristic of solid-liquid phase separation to that characteristic of liquid–liquid phase separation. This problem was discussed in more detail at the end of Section 2.2.4. The difference in the pore size on the surface and in the bulk of the sample associated with the cooling rate gradient was noted in [209]. Work [242] considered the influence of the cooling medium (water) temperature on the porosity and permeability of membranes obtained from mixtures of UHMWPE with PEG and white mineral oil. It was shown that as the temperature of the precipitation bath went up, the pure water flux increased, while the porosity of the membranes passed through a maximum. The observed trends are explained by the authors by taking into account the processes of PEG extraction into water and membrane shrinkage.

#### 2.2.6. Combination of TIPS and Stretching

Despite the fact that the TIPS method, as a way of obtaining porous materials, took shape only in the early 1990s, it was in the early 1980s, researchers began to study the behavior of porous bodies obtained upon cooling of a polymeric solution and subsequent drying of porous bodies upon their drawing [228,243]. However, these papers did not aim to obtain membranes and, hence, study the transport properties of the obtained materials. A combination of the TIPS method and stretching for the preparation of UHMWPE composite membranes was used in [207]. In the cited paper, hollow fibers were obtained using an extruder from a mixture of UHMWPE and mineral oil. Then the solvent was extracted and the obtained hollow fibers were dried and subjected to uniaxial drawing in the draw ratio range from 1 to 6. This paper shows that as the draw ratio becomes bigger, the flux, porosity, bubble point, and tensile strength of the membrane strength increase, while elongation at break decreases. The authors achieved a pure water flux of ~350 l/m^2^ h with a maximum pore size of ~0.45 µm and satisfactory mechanical properties. In a later work [211], this method was used to obtain hybrid membranes consisting of UHMWPE mixtures with PVDF.

Interestingly, stretching can be carried out not only for samples that have already undergone structure formation by TIPS and extraction, as in [207,211,228,243], but also directly in the presence of a solvent. The authors of [35] compared the results of the biaxial drawing of a “gel film” consisting of 70 wt% UHMWPE and 30 wt% LP and “extracted film,” obtained from the same mixture, but with the solvent extracted before drawing. It was shown that when the process was carried out in the presence of a solvent, cavitation nuclei were eliminated, which made the porosity of the resulting membrane dependent on the volume fraction of the solvent in the initial mixture rather than on the draw ratio. As a result, the membranes obtained in the absence of a solvent had higher porosity and ionic conductivity but a wider pore size distribution and worse mechanical properties than those obtained in the presence of a solvent. The extraction of membranes in the presence of a solvent was also carried out in Ref. [47].

Works [192,196] also deserve special attention. In these papers, stretching was carried out not only in the presence of a solvent but also at an elevated temperature. Thus, contrary to the methods discussed above, TIPS proceeded simultaneously with drawing. In these papers, however, little attention was paid to the mechanism of structure formation and it was only stated that an increase in the draw ratio made the pores larger [196], and the porosity passed through an extremum at the temperature of the highest crystallization rate of UHMWPE with an increase in the drawing temperature.

#### 2.2.7. Other Variations of the TIPS Method

Other variations in the TIPS method for the preparation of UHMWPE membranes have also been reported in the literature. In particular, the authors of [225] proposed a method based on the process used to obtain superhydrophobic coatings [244]. The schematic for preparing superhydrophobic UHMWPE/non-woven composite membranes is shown in Figure 9. In the first stage, low-concentration (~0.5 wt% UHMWPE) solutions were prepared in a good solvent (decalin, xylene) at 140 °C. It should be emphasized that these mixtures are characterized by a type II diagram (Figure 4 and Figure 5). Then, a preheated poor solvent (cyclohexanone) was added to the mixture until a good-to-poor solvent ratio of 1:5 was reached. As noted above, as the quality of the solvent deteriorates, the crystallization line only slightly shifts to higher temperatures, in contrast to the liquid equilibrium binodal, which shifts upward sharply. Thus, cyclohexanone addition transforms the phase diagram from type II to type I. In this case, taking into account that the polymer concentration lay to the left of the UCST and was very low, cooling led to the formation of an emulsion of droplets of the polymer-rich phase in the phase depleted by the polymer. Then and after crossing the crystallization line, the polymer crystallized inside the droplets of the polymer-rich phase. After that, the resulting suspension was filtered on a nonwoven substrate and composite membranes were obtained. The properties of the obtained membranes were controlled by choosing the initial solvent, annealing temperature, and amount of filtered suspension. The nanoroughness of the membranes recorded in the work, in our opinion, might be the result of a cosolvent (a solvent–cyclohexanone mixture) release from the amorphous regions of the formed microgels that was not taken into account (see Section 2.2.2).

Another variation of the TIPS method based on the phenomenon of flow-induced crystallization of a low-concentration UHMWPE solution in a good solvent was used in Refs. [226,232,245]. In these works, stirring a solution at a speed of 500–800 rpm at a constant temperature (105 °C) caused winding of spontaneously formed structures of the shish-kebab type on a metal frame or polyethylene terephthalate mesh, and the flow through the obtained membranes reached 35 kg/m^2^ h at an average pore size of ~0.28 µm.

### 2.3. Advantages and Disadvantages of UHMWPE Membrane Formation Methods

Taking into account the data summarized in Section 2, it can be concluded that UHMWPE membranes are nowadays prepared mainly by four methods: by powder sintering, by stretching of monolithic films, by the TIPS method and by the combined method including TIPS and stretching. The advantages and disadvantages of each of the methods, as well as the approximate proportion of the papers devoted to UHMWPE membrane formation by each of the methods, are summarized in Table 2. One can see that almost 70% of the papers and patents (from all the works devoted to UHMWPE membrane formation) deal with the TIPS method (~46%) and its combination with stretching (~23%). Interestingly, an analysis of patent literature shows that, in industry, UHMWPE membranes are generally obtained by the latter method, while no patents report the use of standalone TIPS techniques for UHMWPE membrane formation.

## 3. The Problem of UHMWPE Membrane Shrinkage

Along with the objective difficulties in obtaining homogeneous solutions of UHMWPE, another acute problem is the significant shrinkage of UHMWPE membranes after solvent or extractant removal. For example, it was shown in [209] that the porosity of membranes can drop by more than a factor of two (from 83.4 to 37.4%) as a result of shrinkage. To prevent shrinkage, it is proposed in [206] to fix the membrane in one or two perpendicular directions during the drying procedure. Although this method reduces the effect of shrinkage on porosity, it is noted in this work that membranes fixed in this way can still shrink in one of the directions (thickness). In [35], it was proposed to fix the membranes not only at the drying stage but also at the extraction stage.

The authors of [222] proposed multistage extraction to solve the shrinkage problem. Figure 10 clearly shows how the use of multi-stage extraction (successive replacement of the solvent (mineral oil) with dichloromethane, ethanol and water and subsequent drying from water) can significantly reduce membrane shrinkage compared to single-stage extraction (replacement of the solvent with dichloromethane and subsequent drying). In addition, it is noted in this work that the general trend of the extractant surface tension influence on the shrinkage of the membranes revealed by Matsuyama et al. [246] is violated in the case of UHMWPE membranes. The authors attribute this to the fact that, despite the high value of the surface tension coefficient of water, fewer pores collapse during evaporation because, being highly polar, water does not penetrate small pores in the structure of hydrophobic UHMWPE membranes. It should also be noted that the conclusion made by the authors that the presence of a solvent in the amorphous regions of the polymer can cause shrinkage is in good agreement with the ideas presented in Section 2.2.2 about the mechanism of membrane structure formation by TIPS. A similar method of multistage extraction was also used in Refs. [210,225,232]. In [188], shrinkage was reduced by a one-stage extraction with supercritical carbon dioxide.

It was shown in [47] that composite SiO_2_-containing UHMWPE membranes experience significantly less shrinkage measured at 120 °C (no more than 3%) compared to a membrane without a filler (27 and 30% in the transverse and machine directions, respectively). The authors attribute this increase in thermal stability to the “high thermal resistance of SiO_2_ nanoparticles in the membrane network”. The authors of [52] explain the same effect differently: “nano-SiO_2_ particles were packaged by UHMWPE and HDPE intertwined molecular chains which could reinforce the structural strength of the microporous skeleton and hinder the movement of molecular chains”. The latter explanation, in our opinion, is more correct.

## 4. Modification of UHMWPE Membranes

In some cases, in order to achieve the desired properties of the membranes, after preparation, they are modified to improve their properties or make them hydrophilic/hydrophobic, conductive, etc. Attempts to modify UHMWPE membranes were made in [76,78,79,80,196,247,248,249,250,251]. In particular, in [78,79], the authors carried out plasma treatment of membranes obtained by powder sintering. In [78], the treatment of membranes with cold plasma did not lead to the desired decrease in the water contact angle (increase in hydrophilicity) and even reduced the pure water flux of the membranes. In [79], underwater plasma-assisted pretreatment of microporous UHMWPE membranes in the glow-discharge electrolysis mode was complemented by subsequent plasma-initiated graft copolymerization of the classic monomer acrylic acid, the nonclassic polymer poly(ethylene glycol) and the chemical (bifunctional) crosslinker N,N-methylenebis (acrylamide) (MBA) and this treatment was shown to improve the hydrophilicity and biocompatibility of the resulting product. There are also published data on the increase in the hydrophilicity of UHMWPE membranes by cross-linking with a hydrophilic agent [76] and adding organoclay to the initial mixture [196]. In [80], the modification of membranes with multi-walled carbon nanotubes made it possible to significantly increase the electrical conductivity of the composite membranes. However, the mechanical and transport properties of the modified membranes decreased. In [247], anion-exchange membranes for fuel cells were obtained by chemical modification of ultrathin porous UHMWPE films. In [248], polystyrene was grafted onto a commercially available UHMWPE monolithic film. Then, the grafted polystyrene was sulfonated in the presence of chlorosulfonic and concentrated sulfuric acids, which made it possible to achieve proton conductivity values of 10–60 mS/cm.

## 5. Application of UHMWPE Membranes

Finally, let us discuss the main applications of UHMWPE membranes. First of all, it should be noted that in a number of papers, the authors do not mention any potential applications of the obtained membranes. However, since these papers measure the pure water flux [76,78,188,189,194,197,199,200,201,207,209,210,211,213,215,222,223,225,232], bovine serum albumin rejection [76,194,197,201,209,223], rejection of microspheres [193] or other particles [223] as well as flux recovery ratio [76,194,197,201,213,223] and other characteristics, one can conclude that the membranes are used for liquid media filtration. Some papers [225,232] state that the obtained membranes are designed for direct contact membrane distillation.

The typical value of the pure water flux of UHMWPE membranes is several hundreds of l/m^2^ h bar and that of the rejection rate of bovine serum albumin varies in a wide range, reaching values of 95% and more in the most efficient membranes. Several papers [78,209,222] have reported very high pure water flux values (2000 l/m^2^ h bar and more). However, it is obvious that since the increase in the membrane permeability is generally achieved by increasing the porosity and/or pore size, the membranes with a high pure water flux generally have lower tensile strength and selectivity. In Refs. [76,194,197,201,213,223], the antifouling performance of the UHMWPE membranes was studied. In particular, the flux recovery ratio and other characteristics were measured.

Some papers [123,213,215] also report on the gas flux of the membranes and thus the latter can be used for filtration of gases.

In [35,46,47,51], a combined method of TIPS and stretching was used to prepare UHMWPE membranes to separate the cathode and anode in Li-ion batteries. Table 3 summarizes some properties of the prepared membranes. In particular, one can see from Table 3 that UHMWPE separators have excellent tensile strength (from ~50 to 150 [35,47,51] and even 550 [46] MPa depending on the draw ratio and machine direction). At the same time, the ionic conductivity of the membranes lies within the range from 0.28 to 3.5 mSm/cm depending on the parameters of the membrane formation process. Importantly, the tensile strength values of UHMWPE separators are much more superior to those of the commercially available separators [46]. Studies of the electrochemical performance of the Li-ion batteries with the prepared separators showed that the discharge capacity of the membranes reached 160 mAh/g at a C-rate (reciprocal time needed for full charge in hours) of 0.1 and dropped to ~125 mAh/g with an increase in the C-rate to 8. Furthermore, the discharge capacity dropped by no more than 10% after 50 charge–discharge cycles.

Other possible applications of UHMWPE membranes were discussed in [34,39,193,214,248]. In particular, UHMWPE was used to prepare composite fabrics for personal X-ray shielding [34,39] and effective human body cooling [193], membranes for oil spill recovery [214] and for application in hydrogen-air fuel cells [248].

## 6. Conclusions

An analysis of the literature published over the past 30 years shows that the problem of obtaining membranes from UHMWPE was first mentioned in 1994, but significant interest in this problem increased only ~20 years later (Figure 1). This review provides a brief classification of the methods of polymeric membrane preparation. The progress in the preparation of UHMWPE membranes by methods that do not require the preparation of homogeneous solutions, such as powder sintering, removal of an inert filler and monolithic film drawing, is discussed.

Two points of view (conventional and alternative) of the physicochemical basis of the TIPS method (Section 2.2.1 and Section 2.2.2) are discussed. It is shown that the first concept was developed using incomplete state diagrams of mixtures of pure polymers with a poor and a good solvent and a low MW crystalline substance, whereas in the second concept, phase diagrams were supplemented by a curve reflecting the dependence of the SC polymer swelling degree on temperature. Schemes of phase diagrams for all three types of mixtures are presented and analyzed, and their capacity to predict the thermal behavior of these mixtures and the mechanism of the structural formation of membranes are considered.

Taking into account the conclusions arising from the literature analysis, the review also discusses the problem of preparing homogeneous UHMWPE solutions (Section 2.2.3). Furthermore, a table is given that summarizes the data on the conditions (nature of the low MW liquid, temperature, mixing time, etc.) of the preparation of such solutions.

This review summarizes the results of the studies of thermal behavior in mixtures of UHMWPE with liquid solvents published to date (Section 2.2.4) and presents phase diagrams of UHMWPE mixtures with diphenyl ether, decalin and LP. An analysis of these diagrams shows that there is currently no consensus on what phase diagram type describes thermal behavior, even of the most commonly used mixture of UHMWPE with LP. The review also considers the main methods for controlling the performance properties of membranes obtained by the TIPS method (Section 2.2.5), as well as original variations of the TIPS method for obtaining membranes (Section 2.2.6 and Section 2.2.7). Then, it discusses the problem of shrinkage of UHMWPE membranes and ways to solve it (Section 3), as well as advances in the modification of UHMWPE membranes (Section 4). The main applications of UHMWPE membranes are considered in Section 5.

In light of the foregoing, it can be concluded that the problem of obtaining membranes from UHMWPE is a promising direction that is expected to increasingly attract researchers’ interest in the coming years. From our point of view, the scope of future research is as follows. A further study, in our opinion, is required of the issues related to the development of theoretical foundations of the TIPS method in relation to the production of membranes from UHMWPE: the study of thermodynamics and kinetics of crystallization and liquid decomposition processes, membrane structure formation, and the search for new solvents for UHMWPE. At the same time, given the lack of reports regarding the production of membranes from UHMWPE by the TIPS method from mixtures of this polymer with low MW crystalline substances, such reports will probably be published soon. Considering the fact that UHMWPE is widely used in 3D printing [252,253], it can be expected that the near future will see the development of methods for the preparation of UHMWPE membranes via additive manufacturing [254,255]. In addition, taking into account the difficulties in obtaining homogeneous solutions of UHMWPE, it is possible that methods for obtaining membranes via controlled swelling/deswelling cycles similar to those proposed in [256] will be developed. More studies devoted to the preparation and electrochemical performance of UHMWPE membrane separators are also expected due to the excellent performance of the UHMWPE in this application. Another promising direction is the development of composite membranes with conductive, hydrophilic or superhydrophobic properties.

On the whole, we hope that this review will be useful for researchers already working on the preparation and/or modification of UHMWPE membranes and for a wider circle of “membranologists” interested in the thermodynamics and kinetics of the TIPS process.

## Figures and Tables

**Figure 1 membranes-12-01137-f001:**
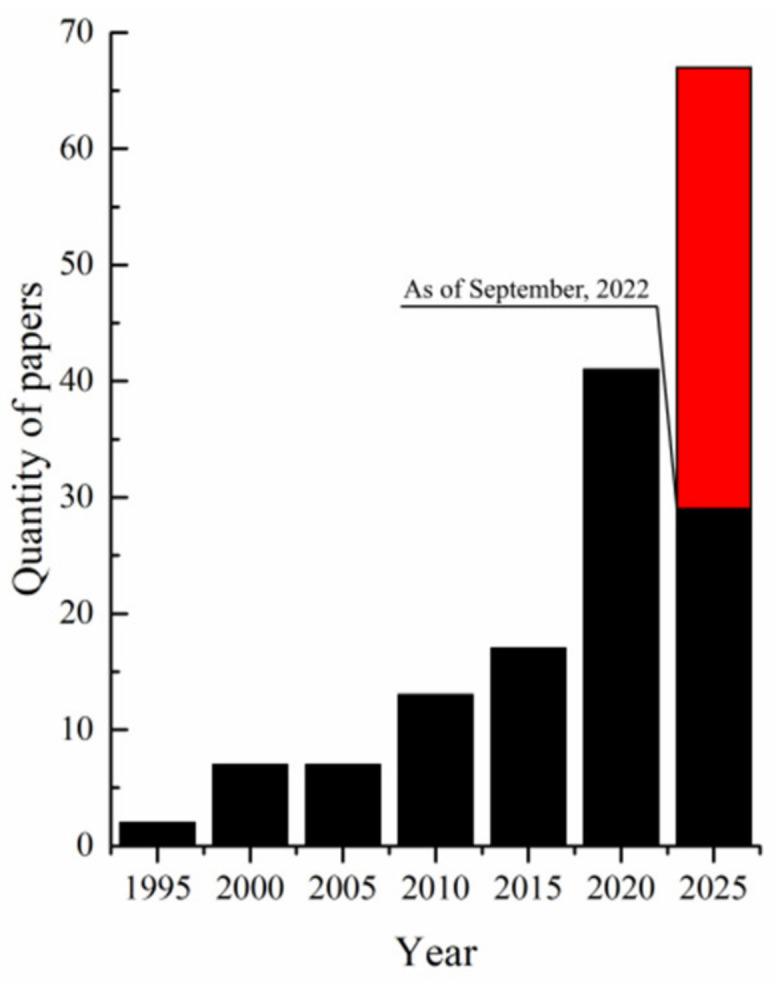
Number of results for “UHMWPE membrane” in quotation marks in Google Scholar. UHMWPE is ultra high molecular weight polyethylene. The horizontal axis represents the years of completion of the five-year interval for calculating the results (1991–1995; 1996–2000; 2001–2005, etc.). The red part shows the expected number of papers by the end of 2025.

**Figure 2 membranes-12-01137-f002:**
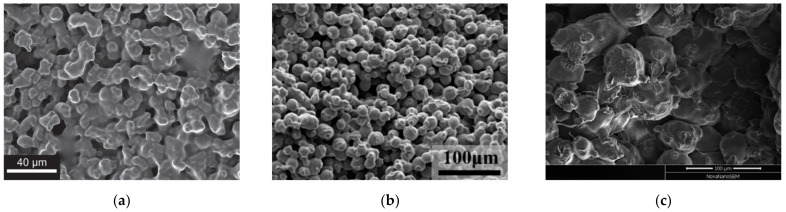
SEM images of the UHMWPE membranes prepared by different variations of the powder sintering method reproduced from [80] (**a**), [81] (**b**) and [76] (**c**). Figure 2a is reproduced with permission from C. Otto, U.A. Handge, P. Georgopanos, O. Aschenbrenner, J. Kerwitz, C. Abetz, A.L. Metze, V. Abetz, Macromolecular Materials and Engineering; Published by “John Wiley and Sons”, 2016. Figure 2c is reproduced with permission from Z. Fei, S. Ying, P. Fab, F. Chen, E. Haque, M. Zhong, Journal of Applied Polymer Science; Published by “John Wiley and Sons”, 2020.

**Figure 3 membranes-12-01137-f003:**
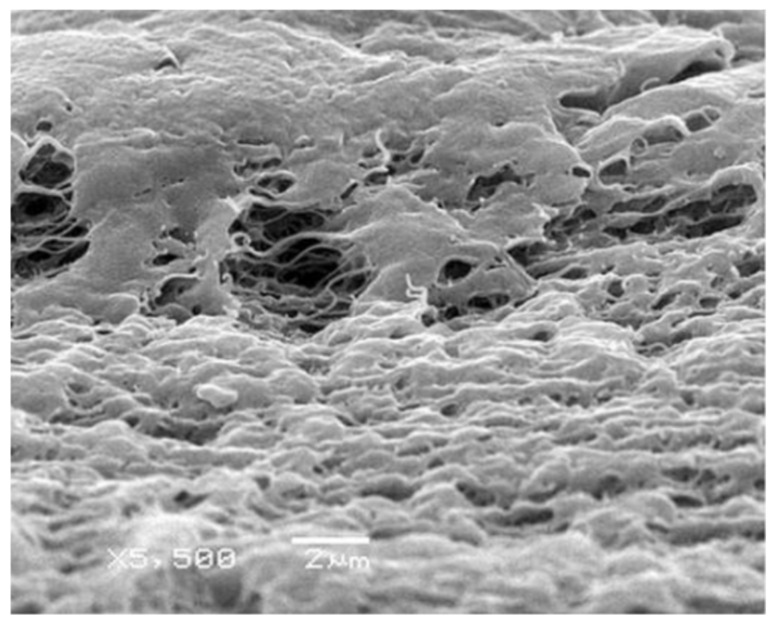
SEM image of the UHMWPE membrane prepared by stretching a monolithic film [123].

**Figure 4 membranes-12-01137-f004:**
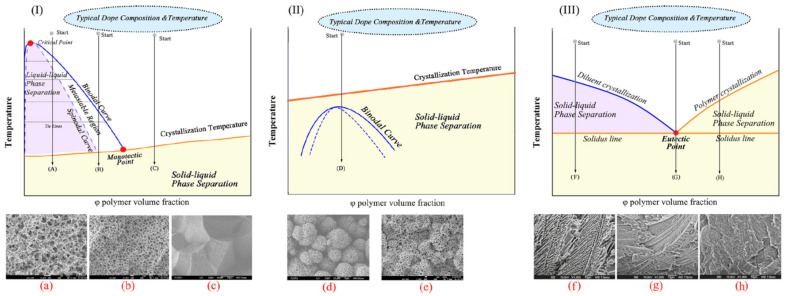
Schematic phase diagrams for the SC polymer mixtures with (**I**) a poor and (**II**) a good liquid solvent and (**III**) eutectic phase diagrams for the SC polymer mixture with a low MW crystalline substance and typical structures formed as a result of the TIPS of the mixtures with different qualitative and quantitative compositions. Subfigures correspond to the typical morphology of capillary-porous bodies formed due to cooling of the homogeneous mixtures by paths (**a**) A, (**b**) B, (**c**) C, (**d**,**e**) D, (**f**) F, (**g**) G and (**h**) H. Phase diagrams I and II and the corresponding SEM images were reproduced with permission from Y. Tang, Y. Lin, W. Ma, X. Wang, Journal of Membrane Science; Published by Elsevier, 2021. SEM images (**f**–**h**) were reproduced with permission from J. Yoon, Alan. J. Lesser, T.J. McCarthy, MACROMOLECULES, 2009, 42, 8827. Copyright year 2009.

**Figure 5 membranes-12-01137-f005:**
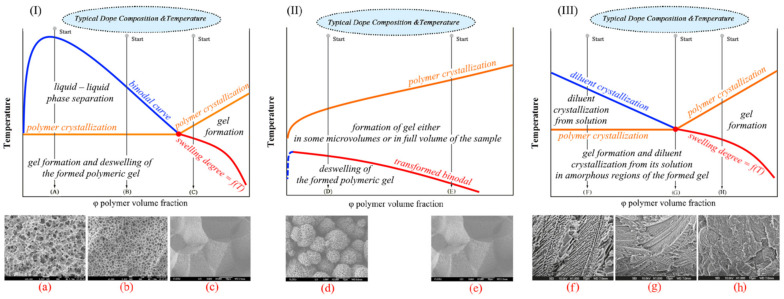
Schematic phase diagrams for the SC polymer mixtures with (**I**) poor and (**II**) good liquid solvents and (**III**) a low MW crystalline substance constructed taking into account the swelling ability of SC polymers and typical structures formed as a result of the TIPS of various mixtures. Subfigures correspond to the typical morphology of capillary-porous bodies formed due to cooling of the homogeneous mixtures by paths (**a**) A, (**b**) B, (**c**) C, (**d**) D, (**e**) E, (**f**) F, (**g**) G and (**h**) H.

**Figure 6 membranes-12-01137-f006:**
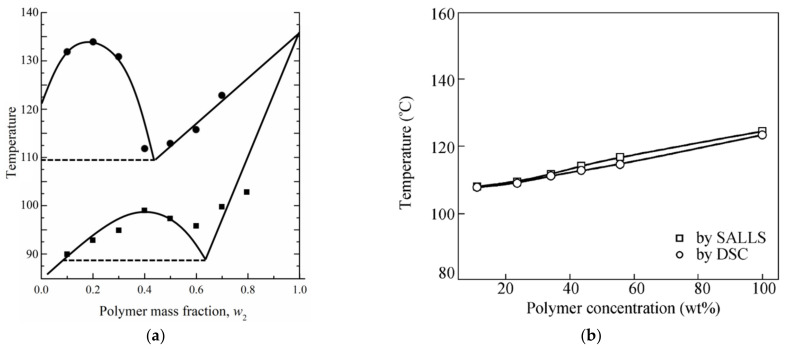
Phase diagrams for UHMWPE mixtures with (**a**) diphenyl ether (circles) and decaline (squares) [230] and (**b**) LP [198]. Figure 6a is reproduced with permission from H. Ding, Y. Tian, L. Wang, B. Liu, Journal of Applied Polymer Science; Published by “John Wiley and Sons”, 2007. Figure 6b is reproduced with permission from L. Shen, M. Peng, F. Qiao, J.L. Zhang, Chinese Journal of Polymer Science, Published by World Scientific, 2008.

**Figure 7 membranes-12-01137-f007:**
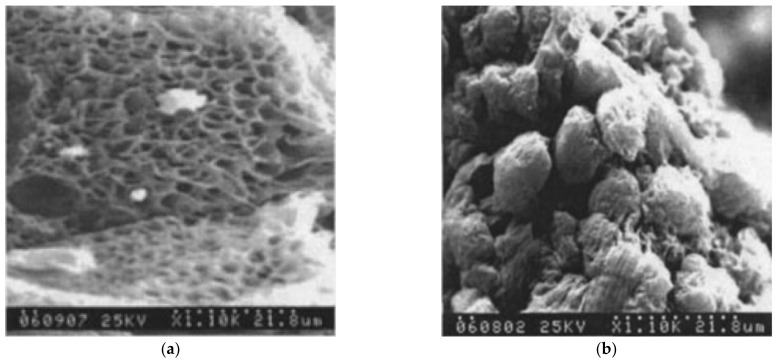
SEM images of capillary-porous bodies prepared by cooling UHMWPE mixtures with (**a**) diphenyl ether and (**b**) decaline containing 40 wt% of the polymer in air. Reproduced with permission from H. Ding, Y. Tian, L. Wang, B. Liu, Journal of Applied Polymer Science; Published by “John Wiley and Sons”, 2007.

**Figure 8 membranes-12-01137-f008:**
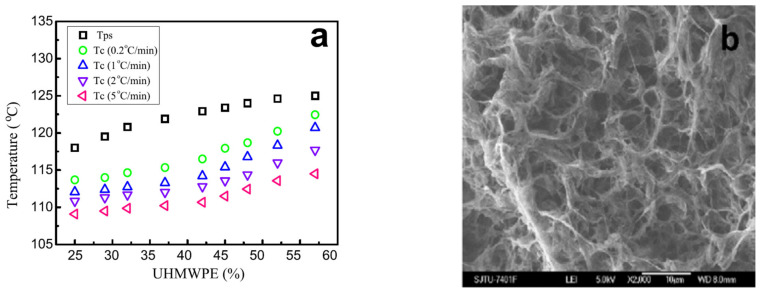
(**a**) Phase diagram for the UHMWPE mixture with LP (the methods used to obtain data are shown in the legend: T_ps_ is the rheological method; T_c_ is the DSC method at the indicated cooling rates) and (**b**) SEM image of the cross-section of the membrane prepared through annealing of the UHMWPE mixture with LP containing 25 wt% of the polymer at 115°C and subsequent cooling. Reproduced with permission from S. Liu, C. Zhou, W. Yu, Journal of Membrane Science; Published by Elsevier, 2011.

**Figure 9 membranes-12-01137-f009:**
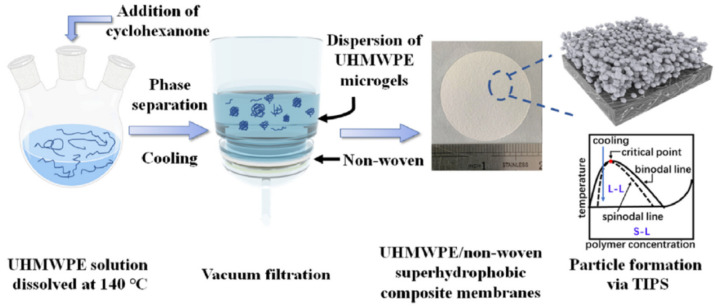
Schematic for preparing superhydrophobic UHMWPE/non-woven composite membranes. Reprinted with permission from J. Quan, J. Yu, Y. Wang, Z. Hu, Journal of Membrane Science, Elsevier, 2022 [225].

**Figure 10 membranes-12-01137-f010:**
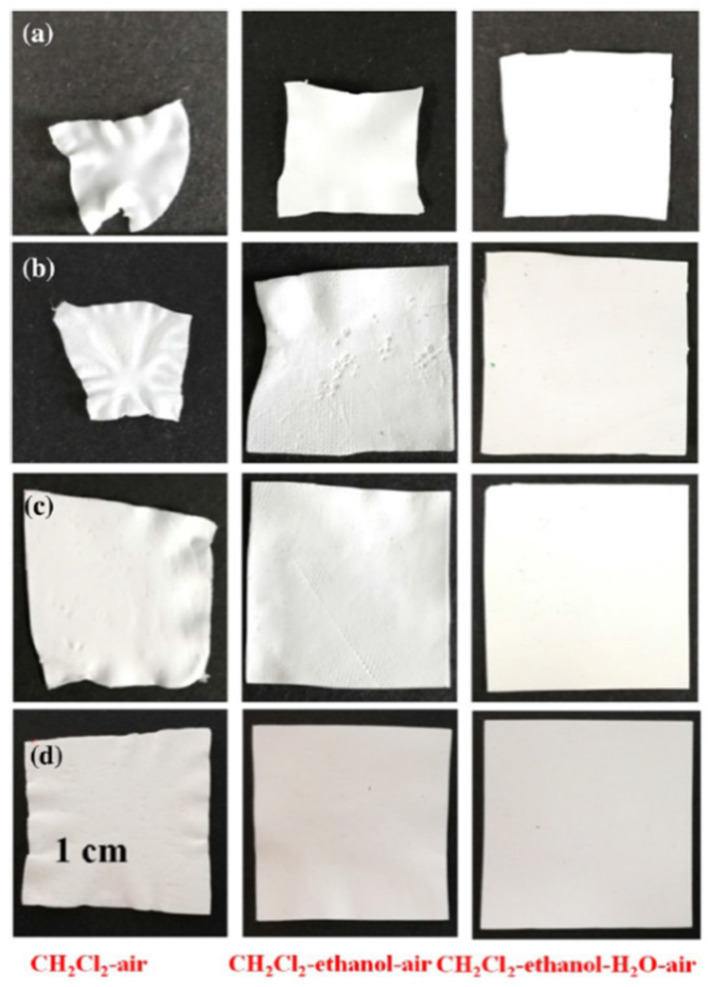
Digital images of the membranes with different annealing times; the membranes were dried by CH_2_Cl_2_-air (left), CH_2_Cl_2_-ethanol-air (middle), and CH_2_Cl_2_-ethanol–water–air (right), respectively; (**a**) is UHMWPE membrane annealing for 0 min, (**b**) is annealing for 5 min, (**c**) is annealing for 15 min, and (**d**) is annealing for 25 min. Reproduced with permission from J. Quan, Q. Song, J. Yu, Y. Wang, J. Zhu, Z. Hu, Advanced Fiber Materials, Published by Springer-Nature, 2022.

**Table 1 membranes-12-01137-t001:** Conditions for preparing homogeneous ultra high molecular weight polyethylene (UHMWPE) solutions.

No.	MW of the UHMWPE Used, 10^6^ g/mol	UHMWPE Concentration in Solution, % wt.	Solvent	Solution Preparation Temperature, °C	Equipment Used, Mixing Conditions	Presence of Antioxidant	Reference
1	2.5	10	Liquid paraffin (LP)	–	rheometer	–	[189]
2	2–4	25	LP	250	Twin screw extruder (TSE)	–	[190]
3	2	10	LP	165	Batch mixer, 100 rpm	–	[191]
4	3.2	1.5	LP	180	–	+	[52]
5	0.5–2	10–50	LP	–	TSE	+	[192]
6	4	5	LP	200	Batch mixer, 60 rpm	Irganox 1076	[193]
7	4	5	LP	200	Batch mixer, 60 rpm	–	[194]
8	2–6	10–60	LP	200	Rheometer, 40 rpm	–	[195]
9	3.7	–	LP	–	TSE	–	[196]
10	4	–	LP	–	–	–	[197]
11	1	70	LP	–	TSE	–	[35]
12	2.5–3.5	23.6–55.6	LP (MW ~500)	160	Rheometer, 60 rpm	1010	[198]
13	1.5–2	10–50	LP (MW ~150–250)	160	Batch mixer, 60 rpm	1% Irganox 1010	[33]
14	2	10	LP (MW ~150–250)	165	Batch mixer, 60 rpm	–	[199]
15	2	10	LP (MW ~150–250)	165	Batch mixer, 80 rpm	–	[200]
16	2.5–4	3–8	LP (MW ~250–450)	200	Batch mixer, 60 rpm	0.7% Irganox 1076	[201]
17	1.2–1.5	10–30.8	LP (MW ~150–300)	190–210	TSE	0.5% Irganox 1010	[202]
18	0.6–5	2–10	LP (MW ~150–300)	190–210	TSE	–	[203]
19	1.2	10–30	LP (MW ~150–300)	185–205	TSE	–	[50]
20	0.5–10	–	light mineral oil	–	–	–	[204]
21	>1	2–20	mineral oil	200	TSE	–	[205]
22	0.4–15	2–30	mineral oil	160	TSE	–	[206]
23	3.65	3.6	Mineral oil	175	–	–	[207]
24	3.65	7.5–15	mineral oil	190	TSE	0.3% Irganox 1076	[208]
25	4.5	10–25	mineral oil	160	torque rheometer	Irganox 1010	[209]
26	3.65	–	Mineral oil 7#	175	–	Irganox 1076	[210]
27	3.65	5–12	Mineral oil 7#	–	TSE	0.3% Irganox 1076	[211]
28	3.65	8	Mineral oil 7#	–	TSE	0.3% Irganox 1076	[212]
29	3.65	5	Mineral oil 7#	180	TSE	0.3% Irganox 1706	[213]
30	3.5	10	mineral oil + petrolatum	200	TSE	0.5% Irganox 1010	[214]
31	3.65	12–24	mineral oil 7# + dibutyl phthalate	–	TSE	0.2% Irganox 1076	[215]
32	9	–	naphtenic oil	165	Batch mixer, 60 rpm	–	[216]
33	>1	10–55	napthenic oil	200	TSE	–	[217]
34	>1	2–20	hydrocarbons	200	TSE	–	[205]
35	0.5–10	–	hydrocarbons	–	–	–	[204]
36	3	1	paraffin	145	–	–	[218]
37	1.6	5	paraffin oil	170		0.1% di-t-butyl-p-cresol	[219]
38	0.4–15	2–30	paraffin oil	160	TSE	–	[206]
39	>1	10–55	paraffin oil	200	TSE	–	[217]
40	5	4–5	paraffin oil	150	–	–	[220]
41	4	2–3.2	paraffin oil	150	Batch mixer	–	[221]
42	1.6	5	paraffin oil #70	220	TSE, 60 rpm	0.7% Irganox 1076	[34]
43	0.6	20–25	paraffin oil	220	TSE	+	[47]
44	2.7	4–12	paraffin oil	150–250	TSE	–	[111]
45	–	–	paraffin oil	–	TSE	–	[51]
46	1.6	5	paraffin oil #70	220	TSE	0.7% Irganox 1076	[222]
47	>1	2–20	paraffin wax	200	TSE	–	[205]
48	1.6	20	white oil #70		TSE	0.7% Irganox 1076	[223]
49	5	2–5	hexadecane	135	–	–	[43]
50	5	50	hexadecane	160–200	Twin screw microcompounder, 100 rpm	0.2% Irganox 1010 + 0.2% Irgafox 168	[224]
51	>1	10–55	aromatic oil	200	TSE	–	[217]
52	4	2–3.2	paraffin oil	150	Batch mixer	–	[221]
53	2–6	10–60	polybutene-1 (MW ~400)	200	Rheometer, 40 rpm	–	[195]
54	1.6–9	0.5	mixture of xylenes	140	–	–	[225]
55	2–3	0.058	xylene	140	–	–	[226]
56	1	38	p-xylene	140	TSE	–	[227]
57	2	0.25–0.75	p-xylene	130	–	–	[44]
58	5	2–5	o-xylene	135	–	–	[43]
59	1.6–9	0.5	o-xylene	140	–	–	[225]
60	3.5	0.6–10	decalin	160	–	0.5% di-t-butyl-p-cresol	[228]
61	0.4–15	2–30	decalin	160	TSE	–	[206]
62	1.6	5	decalin	150		0.1% di-t-butyl-p-cresol	[219]
63	6	0.4	decalin	135	–	–	[229]
64	3	10–40	decalin	160	–	–	[230]
65	3	1	decalin	145	–	–	[218]
66	4	2–3.2	decalin	150	Batch mixer	–	[221]
67	1.6	5–15	decalin	165	TSE, 60 rpm	0.7% Irganox 1076	[39]
68	5	50	decalin	160–200	Twin screw microcompounder, 100 rpm	0.2% Irganox 1010 + 0.2% Irgafox 168	[224]
69	5	50	decalin + dodecanol	160–200	Twin screw microcompounder, 100 rpm	0.2% Irganox 1010 + 0.2% Irgafox 168	[224]
70	2.2–4.4	0.5–4	Decalin + cyclohexanone	130	–	–	[231]
71	3	0.07	decalin	140	–	1076	[232]
72	1.6–9	0.5	decalin	140	–	–	[225]
73	0.4–15	2–30	tetralin	160	TSE	–	[206]
74	3	10–40	diphenyl ether	160	–	–	[230]
75	>1	2–20	dioctyl phthalate	200	TSE	–	[205]
76	>1	10–55	dioctyl phthalate	200	TSE	–	[217]
77	34	0.9–3	dioctyl phthalate	220	–	–	[233]
78	>1	10–55	butylbenzyl phthalate	200	TSE	–	[217]
79	>1	2–20	dibutyl sebacate	200	TSE	–	[205]
80	>1	10–55	dibutyl phthalate	200	TSE	–	[217]
81	1	5	1,2-dichloroethane + cyclopentane	180	Autoclave with CO_2_, 5 MPa	Irganox 1010	[234]
82	1.2–6	4	1,2,4-trichlorobenzene	–	–	–	[45]
83	5	50	stearic acid	160–200	Twin screw microcompounder, 100 rpm	0.2% Irganox 1010 + 0.2% Irgafox 168	[224]

**Table 2 membranes-12-01137-t002:** Conditions for preparing homogeneous UHMWPE solutions.

Method of UHMWPE Membrane Preparation	Advantages	Disadvantages	Proportion of Papers Devoted to the Formation of Membranes by the Respective Method
Powder sintering	No shrinkage, simplicity of hardware design, no need to prepare a homogeneos solution, high tensile strength	Low porosity, wide pore size distribution, low selectivity, difficulty in control of the porous structure and properties	~18
Stretching	No need to prepare a homogeneous solution, high tensile strength	High thermal shrinkage, difficulty in control of the porous structure and properties	~5
TIPS	High porosity, good possibilities of control of the porous structure and properties, possibility for modification by inclusion of fillers in the dope solution	Problem of homogeneous solution preparation, lower tensile strength, high shrinkage, low surface porosity	~46
Combination of TIPS and stretching	High porosity, excellent possibilities of control of the porous structure and properties, possibility for modification by inclusion of fillers in the dope solution, high tensile strength	Problem of homogeneous solution preparation, high shrinkage	~23
Other	–	–	~8

**Table 3 membranes-12-01137-t003:** Conditions for preparing homogeneous UHMWPE solutions.

Property	Reference			
	[47]	[35]	[51]	[46]
Tensile strength, MPa	56–162	58–130	52–145	180–550
Ionic conductivity, mSm/cm	1.5–3.5	0.4–3.2	0.28–1.52	–
Discharge capacity at a C-rate of 1, mAh/g	147	131	148	152

## Data Availability

Not applicable.

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
