# Peer review of "Current State-of-the-Art in Membrane Formation from Ultra-High Molecular Weight Polyethylene"

_membranes, 2022, doi:10.3390/membranes12111137_

Round 1

Reviewer 1 Report

The article entitled “Current state-of-the-art in membrane formation from ultra-high 2 molecular weight polyethylene” presents the comprehensive review of polyethylene membrane fabrication. The article is well written, however minor revision is needed before this paper can be considered for the publication.

1.      Scope of article can be concluded in abstract section.

2.      Citation could be significantly reduced in Introduction section. The significance of polyethylene material over other material could be discussed with respect to hydrophilic properties.

3.      In Section 2, SC can be elaborated in line no.59.

4.      Advantages and disadvantage of each polyethylene membranes fabrication methods could be compared and tabulated.

5.      Application of polyethylene membrane can be elaborate in detail with respect to permeability and solute rejection.

6.      Future scope of research could be elaborated in detail.

Author Response

Response to Reviewer 1 Comments

  1. Scope of article can be concluded in abstract section.

Response 1. Thank you for the suggestion. We have added the scope of the article to the abstract.

  1. Citation could be significantly reduced in Introduction section. The significance of polyethylene material over other material could be discussed with respect to hydrophilic properties.

Response 2. Thank you for the comment. Please note that most of the papers cited in the first section were not for introductory purposes only. The data from these references is discussed later in section 2.2.1., 2.2.3., 4., etc. We have also added some information on relevant (in terms of membrane formation) properties of polyethylene in lines 39-53.

  1. In Section 2, SC can be elaborated in line no.59.

Response 3. The abbreviation SC stands for semicrystalline. We have added the definition to line 77.

  1. Advantages and disadvantage of each polyethylene membranes fabrication methods could be compared and tabulated.

Response 4. Thank you for the valuable suggestion. We have introduced a new section “Comparison of UHMWPE membrane formation methods” where we have added a table (2) and corresponding discussion.

  1. Application of polyethylene membrane can be elaborate in detail with respect to permeability and solute rejection.

Response 5. Thank you for the valuable suggestion. We have added one more section “5. Application of UHMWPE membranes” with the corresponding discussion.

  1. Future scope of research could be elaborated in detail.

Response 6. Thank you for the comment. Some information on the prospects of the development in the field is given in the conclusion in lines 990–1009. However, we have expanded this discussion a little, following the reviewer`s suggestion.

Reviewer 2 Report

This manuscript provides a thorough overview of the fabrication methods for ultra-high molecular weight polyethylene membranes, which are classified into solvent-free and solvent-based groups. In particular, a detailed fundamental discussion was given of the phase diagrams of the UHMWPE solution to explain the morphology of the resultant membranes in the TIPS method, which would be very beneficial to the membrane community.  While, some contents, as commented below, are missing in the current version, which could fill the overall picture of this topic:

1.       Please provide the intended applications of the UHMWPE membrane, and thus the expected properties, like porosity, pore size, retention rate, etc., of the UHMWPE membranes.

2.       Please provide the basic overview of the physical properties of UHMWP, which will affect the membrane fabrication process, prior to Section 2.

3.       PE membranes have been maturely fabricated and employed in the market, so why do we want to develop UHMWPE membranes, especially with so many unsolved issues?

4.       Please move Figure 10 to section 1.

5.       Line 69, which group the “second group” is referred to, the solvent-based or solvent-free method?

6.       The current manuscript introduced the “solvent-free” methods very briefly (roughly half a page in total). Is this because the portion of these methods is very little? If yes, can the authors provide information about the shares of “solvent-free” and TIPS methods for UHMWP membrane fabrication?

7.       Would authors briefly introduce the TIPS process to readers before entering the detailed discussion?

8.       In section 2.2.1, the authors mentioned the binodal curve in the type II phase diagram was discussed in the 70s, but has rarely been mentioned since the 90s (line 251 to line 258). It seems to readers that this concept has not been employed in recent studies, and the readers may wonder why the authors want to include it in a review mainly focusing on the progress from the 90s to the present.

9.       In line 538, the authors suggested the porous morphology shown in figure 6a resulted from liquid-liquid phase separation followed by polymer crystallization, which is the typical path of type I. However, the authors believe the corresponding phase diagram is the type II (532), which is very confusing to readers.

10.   Section 2.2.2 indeed provides meaningful discussion to the readers. However, it seems most of the related content is the authors’ work. Have other researchers proposed or explained the disagreement between traditional phase diagrams and resultant morphology?

11.   In line 722, the authors claim that the stepwise cooling with annealing was disused at the end of section 2.2.4, which seems to be absent.

12.   What does the “membrane strength” in line 741 mean?

13.   The last paragraph on page 25, “the modification of membrane with MWCT increases the electrical conductivity due to some decrease in the mechanical and transport properties” what does this mean?

14.   Some numberings are very confusing

A)      In figure 3, since the letter e, figure “x” is not responding to path “x”.

B)      The “figure 3” in line 320 is referred to figure 3 or figure 4?

C)      Figure 4. e is the same as Fig. 4C and different with the fig. 3e. Is it supposed to be this way?

15.   Some typos need to be corrected:

A)      Line 87, “for fine control of the structure the structure of the obtained …”

B)      Line 156-159, “Monolithic UHMWPE films used as the starting material in membrane formation via stretching are obtained by hot pressing of the powders at a temperature lower or higher than the polymer melting temperature [91,117–120], by calendaring [121], hot pressing, skiving off the block [122], etc”. The “hot pressing” is mentioned twice in the enumeration.

Author Response

Response to Reviewer 2 Comments

  1. Please provide the intended applications of the UHMWPE membrane, and thus the expected properties, like porosity, pore size, retention rate, etc., of the UHMWPE membranes.

Response 1. Thank you for the valuable suggestion. We have added one more section “5. Application of UHMWPE membranes” with the corresponding discussion.

  1. Please provide the basic overview of the physical properties of UHMWP, which will affect the membrane fabrication process, prior to Section 2.

Response 2. Thank you for the suggestion. We have expanded the discussion of UHMWPE properties by adding another paragraph in lines 39–53.

  1. PE membranes have been maturely fabricated and employed in the market, so why do we want to develop UHMWPE membranes, especially with so many unsolved issues?

Response 3. Thank you for the question. In our opinion, one reason for that is that scientists are always in continuous search for new materials for membrane preparation and UHMWPE is one of many less common choices. The other is that UHMWPE membranes, if the difficulties in their preparation have been overcome, have some significant advantages compared to standard PE membranes. In particular, they have much higher mechanical strength and abrasion resistance, which is important in various applications.

  1. Please move Figure 10 to section 1.

Response 4. We have moved figure 10 and the corresponding discussion from the conclusion to the introduction.

  1. Line 69, which group the “second group” is referred to, the solvent-based or solvent-free method?

Response 5. Thank you for the comment. Line 69 referred to the solvent-based methods. The mistake has been corrected.

  1. The current manuscript introduced the “solvent-free” methods very briefly (roughly half a page in total). Is this because the portion of these methods is very little? If yes, can the authors provide information about the shares of “solvent-free” and TIPS methods for UHMWP membrane fabrication?

Response 6. Thank you for the comment. In fact the discussion of solvent-free methods is almost two pages long. However, we agree with the reviewer that it is quite brief in comparison with solvent-based methods. You are right in the assumption that the portion of the solvent-free methods for UHMWPE membrane formation is quite small. We have summarized the approximate portions of different methods used in academic papers and patents to prepare UHMWPE membranes in Table 2.

  1. Would authors briefly introduce the TIPS process to readers before entering the detailed discussion?

Response 7. Some most general information about the TIPS method is given in lines 193–201. We expanded this discussion a little.

  1. In section 2.2.1, the authors mentioned the binodal curve in the type II phase diagram was discussed in the 70s, but has rarely been mentioned since the 90s (line 251 to line 258). It seems to readers that this concept has not been employed in recent studies, and the readers may wonder why the authors want to include it in a review mainly focusing on the progress from the 90s to the present.

Response 8. Thank you for the comment. The review is indeed focused on the progress from the 1990s to the present time since UHMWPE was not used as membrane material in the earlier periods. However, the development of the fundamentals of the TIPS method, phase diagrams for the semicrystalline polymer mixtures with low MW substances and, more generally, thermal behavior in such mixtures started in the late 1930s. These earlier data that are not directly connected to the UHMWPE membranes can, in our opinion, be significant for the discussion. It should also be noted that Figure 3 is reprinted from another recent review [ref. 20 in the manuscript] (this figure also appears in other recent papers, such as [132]). Thus, the concepts developed as early as in the 1970s are still used today in theoretical speculations. It is surprising in our opinion that the curve that, in fact, was never constructed experimentally is used so widely to discuss membrane formation mechanisms and it is one of the reasons why we believe that the conventional physico-chemical basis of the TIPS method should be revised and improved.

  1. In line 538, the authors suggested the porous morphology shown in figure 6a resulted from liquid-liquid phase separation followed by polymer crystallization, which is the typical path of type I. However, the authors believe the corresponding phase diagram is the type II (532), which is very confusing to readers.

Response 9. Thank you for the comment. We have rewritten the corresponding discussion to avoid the confusion. The problem was that Figure 5a contains two phase diagrams for different mixtures (UHMWPE + decalin and UHMWPE + diphenyl ether). Although the authors of the original paper classify both phase diagrams as type I, in our opinion, based on the analysis of SEM images (Figure 6) only the phase diagram for the UHMWPE + diphenyl ether mixture (circles on Figure 5a) corresponds to type I, while the phase diagram for the UHMWPE + decalin system (squares in Figure 5a) corresponds to type II.

  1. Section 2.2.2 indeed provides meaningful discussion to the readers. However, it seems most of the related content is the authors’ work. Have other researchers proposed or explained the disagreement between traditional phase diagrams and resultant morphology?

Response 10. Thank you for the comment. Indeed most of the published research devoted to the membrane formation by the TIPS method is more or less in line with the conventional understanding presented in section 2.2.1. However, some cited references contribute to our point of view (for example, 165, 171,172). We have also added some new references which are (at least in some parts) are in line with the alternative understanding of the TIPS process [147, 174-176, 179-183]. Some arguments including those proposed by other researchers are discussed in our experimental papers (for example, see page 4 in [159]). However, since the present review is devoted to UHMWPE membranes (although with focus on TIPS), rather than the phase equilibria in SC polymer – low MW substance systems, we feel that it would not be appropriate to include such quite lengthy discussion in the review. It should also be noted that although, for example, mention of small pores inside polymer spherulites was found in several papers published by other authors [174–176,179–183], no attempts (except for our works, as far as we are aware) have been made to propose a mechanism of their formation. In our opinion, the formation of such pores is inexplicable using the conventional basis of the TIPS method and thus the authors remain silent on the matter.

  1. In line 722, the authors claim that the stepwise cooling with annealing was disused at the end of section 2.2.4, which seems to be absent.

Response 11. Thank you for the remark. The sentence in line 722 has been rewritten.

  1. What does the “membrane strength” in line 741 mean?

Response 12. We meant “tensile strength of the membranes”. The mistake has been corrected.

  1. The last paragraph on page 25, “the modification of membrane with MWCT increases the electrical conductivity due to some decrease in the mechanical and transport properties” what does this mean?

Response 13. We have rewritten the sentence for better readability.

  1. Some numberings are very confusing
    A) In figure 3, since the letter e, figure “x” is not responding to path “x”.

Response 14A. Although this is an “artifact” of the paper from which the corresponding figure was reprinted we have changed the numeration for better readability.

B) The “figure 3” in line 320 is referred to figure 3 or figure 4?

Response 14B. We have removed the parenthesis from the sentence. “Figure 3” is correct.

C) Figure 4. e is the same as Fig. 4C and different with the fig. 3e. Is it supposed to be this way?

Response 14C. Thank you for the remark. Yes, since the mechanism of structure formation in the polymer-rich mixtures of the SC polymer with a poor and a good solvent is the same, the structures prepared from such mixtures are very similar, so the figure is correct.

  1. Some typos need to be corrected:
    A) Line 87, “for fine control of the structure the structure of the obtained …”
    B) Line 156-159, “Monolithic UHMWPE films used as the starting material in membrane formation via stretching are obtained by hot pressing of the powders at a temperature lower or higher than the polymer melting temperature [91,117–120], by calendaring [121], hot pressing, skiving off the block [122], etc”. The “hot pressing” is mentioned twice in the enumeration.

Response 15. Thank you for the remarks, the duplicates have been removed.

Reviewer 3 Report

The manuscript titled "Current state-of-the-art in membrane formation from ultra-high molecular weight polyethylene" discusses in detail the use of ultrahigh molecular weight polyethylene (UHMWPE) for the synthesis of membranes. The manuscript collectively presents the research carried out in the field of membrane science for the synthesis of UHMWPE-based membranes using the TIPS method taking into consideration the phase diagrams. The manuscript is well arranged and in line with the membranes journal.

Author Response

There are no questions/suggestions in the reviewer 3 report.

Round 2

Reviewer 2 Report

The current form is ready for publishing.